# Physiologic Effects of Instilled and Aerosolized Surfactant Using a Breath-Synchronized Nebulizer on Surfactant-Deficient Rabbits [note 1]

**DOI:** 10.3390/pharmaceutics13101580

**Published:** 2021-09-29

**Authors:** Robert M. DiBlasi, Kellie J. Micheletti, Joseph D. Zimmerman, Jonathan A. Poli, James B. Fink, Masaki Kajimoto

**Affiliations:** 1Department of Respiratory Care, Seattle Children’s Hospital, Seattle, WA 98105, USA; kellie.micheletti@seattlechildrens.org (K.J.M.); Joseph.Zimmerman@seattlechildrens.org (J.D.Z.); 2Center for Integrative Brain Research, Seattle Children’s Research Institute, Seattle, WA 98101, USA; Jonathan.Poli@seattlechildrens.org (J.A.P.); masaki.kajimoto@seattlechildrens.org (M.K.); 3Aerogen Pharma Corporation, San Mateo, CA 94402, USA; jfink@aerogenpharma.com; 4Division of Respiratory Care, Department of Cardiopulmonary Sciences, Rush University Medical Center, Chicago, IL 60612, USA

**Keywords:** aerosol, surfactant, RDS, nebulizers

## Abstract

Surfactant administration incorporates liquid bolus instillation via endotracheal tube catheter and use of a mechanical ventilator. Aerosolized surfactant has generated interest and conflicting data related to dose requirements and efficacy. We hypothesized that aerosolized surfactant with a novel breath-actuated vibrating mesh nebulizer would have similar efficacy and safety as instilled surfactant. Juvenile rabbits (1.50 ± 0.20 kg, *n* = 17) were sedated, anesthetized, intubated, and surfactant was depleted via lung lavage on mechanical ventilation. Subjects were randomized to receive standard dose liquid instillation via catheter (*n* = 5); low dose surfactant (*n* = 5) and standard dose surfactant (*n* = 5) via aerosol; and descriptive controls (no treatment, *n* = 2). Peridosing events, disease severity and gas exchange, were recorded every 30 min for 3 h following surfactant administration. Direct-Instillation group had higher incidence for peridosing events than aerosol. Standard dose liquid and aerosol groups had greater PaO_2_ from pre-treatment baseline following surfactant (*p* < 0.05) with greater ventilation efficiency with aerosol (*p* < 0.05). Our study showed similar improvement in oxygenation response with greater ventilation efficiency with aerosol than liquid bolus administration at the same dose with fewer peridosing events. Breath-synchronized aerosol via nebulizer has potential as a safe, effective, and economical alternative to bolus liquid surfactant instillation.

## 1. Introduction

Premature infants are commonly born with critically underdeveloped, surfactant-deficient lungs. Surfactant-deficiency is associated with onset of respiratory distress syndrome (RDS), a major cause of morbidity and mortality in premature infants worldwide. Endogenous surfactant is a biochemical compound that forms a layer between the terminal airways/alveolar surfaces and alveolar gas. Surfactant reduces surface tension, improves lung compliance, and stabilizes lung volumes at a low transpulmonary pressure [1]. Surfactant-deficiency can lead to alveoli that might never inflate or can collapse on expiration and require excessive effort to re-expand on inspiration, leading to the development of severe respiratory failure, air leak syndromes, and death [2].

Intratracheal administration or ‘direct-instillation’ of exogenous liquid surfactant via an endotracheal tube (ETT) or catheter is the approved ‘gold standard’ for surfactant replacement therapy (SRT) in infants. This revolutionary therapy contributed to significant reductions in infant mortality [3,4]. However, direct-instillation of exogenous surfactant can transiently occlude the airways and exacerbate respiratory failure. As such, infants may require prolonged endotracheal intubation and mechanical ventilator support in case clinical deterioration occurs. Common procedural complications, or ‘peridosing events’, include coughing, gagging, plugging of the ETT, hypercarbia, refractory hypoxemia, bradycardia due to hypoxia, tachycardia due to agitation, and changes in arterial blood pressure (BP) [1]. Furthermore, direct-instillation efficacy can be limited by gravity dependent delivery of surfactant to only one lung, resulting in variable and suboptimal dosing [5]. In the period immediately following surfactant dosing, ventilator settings may need to be increased to prevent reduced gas delivery to the lungs (atelectotrauma) [6], clinical deterioration, and poor gas exchange [7]. As lung mechanics improve over the minutes and hours following therapy, failure to wean ventilator settings may result in excessive gas delivery to the lungs (volutrauma) [5]. These confounding factors contribute to prolonged need for invasive mechanical ventilation, increased pulmonary injury/inflammation, deactivation of endogenous surfactant production, reduced therapeutic effect of surfactant, pulmonary growth arrest and increased incidence of chronic lung disease such as bronchopulmonary dysplasia [2,8,9]. Additionally, acute changes in lung mechanics following therapy not only puts premature infants at risk for ventilator-induced lung injury (VILI) but could cause arterial carbon dioxide (PaCO_2_) levels to change [5]. This in turn could contribute to fluctuations in cerebral vascular tone and perfusion, and development of serious neurologic complications [10,11,12,13]. Animal studies have shown that the distribution of liquid surfactant from direct-instillation is often non-uniform, and that non-uniform distribution patterns are associated with a poorer clinical response [14,15]. Nebulized or ‘aerosolized’ surfactant involves generating small surfactant droplets (~1–5 µm) that can be inhaled into the lungs more easily than liquid bolus without altering the functional properties of surfactant [16,17]. Aerosolization of surfactant may result in more uniform distribution of drug in the lungs [18], with fewer adverse effects on airway obstruction, hemodynamics, cerebral blood flow, and gas exchange than does direct-instillation [19]. Treatment with aerosolized surfactant has been attempted in bench, animal and clinical trials [20]. All studies utilized continuous nebulization and while finding of safety where consistent reports of efficacy were limited. Even with minimal pulmonary deposition, signals of clinical efficacy were observed after large doses of Alveofact were administered [21]. As such, greater delivery efficiency would be required to make aerosol delivery of surfactant feasible, suggesting potential benefit of breath synchronization of small aerosol particles.

A recent development in nebulizer technology is a prototype breath-synchronized small particle vibrating mesh nebulizer (VMN) (Aerogen Pharma, San Mateo, CA, USA) that has low deadspace and is placed in series between the patient-Y connector of the ventilator circuit and ETT. Nebulized surfactant synchronized with inhalation could deliver more efficient pulmonary drug deposition than direct-instillation and improve clinical response with fewer adverse peridosing events. However, there are limited pre-clinical data to support appropriate dosing, device performance, safety, and efficacy for surfactant delivery with a breath synchronized VMN in neonates.

We hypothesized that aerosolization of a bovine surfactant using a novel breath-actuated VMN would have similar efficacy and safety to a similar dose of instilled surfactant in an intubated mechanically ventilated, surfactant-deficient, lung injured animal model. In addition to standard treatment doses used with aerosol and direct-instillation, we nebulized 1/2 standard dose in one group to establish dosing between small particles generated with aerosol and the relatively large liquid particles with direct-instillation. 

## 2. Methods

A randomized controlled animal study was designed to compare pre-clinical physiologic outcomes on respiratory function and peridosing events between direct-installation and aerosolized surfactant. Sample size calculation for surfactant treatment (*n* = 5 per group) was based on power analysis using mean and standard deviation of partial pressure of oxygen in the arterial blood (PaO_2_) from a previous rabbit study that compared nebulized to directly instilled surfactant in intubated mechanically ventilated rabbits [22]. We included descriptive control animals (*n* = 2) that did not receive surfactant, based on our previous work with a similar animal rabbit injury model where disease severity was consistent up to 4 h following lung lavage [23]. Final sample size was determined as five animals per treatment group and two descriptive controls. Figure 1 shows the experimental design and procedures. 

### 2.1. Surgical Preparation/Instrumentation 

All experimental animal procedures were conducted according to the National Institute of Health’s Guide for the Care and Use of Laboratory Animals and approved by Seattle Children’s Institutional Animal Care and Use Committee. Juvenile female New Zealand White rabbits (Western Oregon Rabbit Company, Philomath, OR, USA), were tranquilized with 1 mg/kg acepromazine and anesthetized with 33 mg/kg ketamine and 6.6 mg/kg intramuscular xylazine. The neck and upper chest were shaved. Cetacaine spray (0.1%) was applied topically to the oral pharynx and glottis to reduce gag reflex and laryngospasm during oral intubation. Local anesthesia was provided around the trachea with lidocaine. The trachea was dissected, isolated and animals were orally intubated with a 2.5 mm ID ETT under direct laryngoscopy. Following endotracheal intubation, a strand of umbilical tape was used to anchor the trachea and ETT, to prevent displacement, gas leakage, and saline leakage during lavage.

Animals were initially ventilated (Draeger, Babylog VN500, Lubeck Germany) with volume guarantee in the assist/control mode with fraction of inspired oxygen (FiO_2_) of 0.5, minimum respiratory rate of 40 breaths/min, a tidal volume (V_T_) of 6 mL/kg, inspiratory time of 0.30 s, and a positive end-expiratory pressure (PEEP) of 5 cmH_2_O. The respiratory rate was adjusted initially to maintain arterial blood pH of 7.35–7.45 and remained at this setting throughout the remainder of the study. Animals were not administered neuromuscular blocking agents and could spontaneously trigger mandatory ventilator breaths above the set respiratory rate consistent with clinical management of pre-term infants. The oxygen saturation (SpO_2_) was monitored (Rad 7, Massimo, Italy) with a tail probe and body temperature was monitored continuously with a rectal temperature probe and maintained normothermic (38–39 °C) using a warming pad. A 20-gauge angiocatheter was placed in the right jugular vein for administration of fluids and medications. Sedation and analgesia were maintained to minimize pain and promote spontaneous breathing efforts using continuous intravenous infusion of ketamine and xylazine (3 and 0.18 mg·kg^−1^·hr^−1^, respectively) and titrated based on blood pressure, heart rate, and maintenance of anesthetic plane. Maintenance intravenous fluids were provided with continuous infusion at 3 mL·kg^−1^·hr^−1^ of 0.9% saline containing 5% dextrose. A 22-gauge angiocatheter was placed in the right carotid artery for heart rate and BP monitoring and sampling for arterial blood-gas analyses (ABGs). ABGs (pH, PaCO_2_, PaO_2_) were measured at regular intervals using a Radiometer ABL 800 (Radiometer America, Brea, CA, USA).

A 6 FR esophageal balloon catheter (Cardinal Healthcare, Dublin, OH, USA) was positioned in the lower esophagus to measure esophageal pressure (P_ES_) and estimate changes in pleural pressures and effort of breathing [24]. The chest was shaved, and sixteen ECG electrodes were applied circumferentially around the chest and back at the level of the 5th intercostal space using a small amount of electrode gel. Electrodes were attached to the chest to estimate end-expiratory cross-sectional distribution of ventilation based on electrical impedance tomography (EIT; Pulmovista 500 system, Draeger, Lubeck, Germany). A calibrated proximal hot-wire flow sensor positioned between the ventilator circuit patient Y and ETT provided real-time measurements of respiratory rate, airway pressure, flow, and V_T_. All data were recorded for 45 s at 1040 Hz at each timepoint.

Gases were conditioned to 37–40 °C with a heated humidifier (MR 850, Fisher Paykel Healthcare, Auckland, New Zealand) and heated-wire ventilator circuits. The expiratory limb of the ventilator circuit was filtered with two high-efficiency particulate air (HEPA) filters (Draeger, Lubeck, Germany) placed in series to prevent exhaled aerosol from entering and compromising the expiratory valve. Prior to lung lavage, ABGs, P_ES_, lung mechanics, ventilation parameters, and EIT measurements were obtained. 

### 2.2. Induction of Surfactant-Deficiency and Lung Injury 

The FiO_2_ was increased to 1.0, and the lungs were depleted of surfactant and injured by repeated lavage with 25 mL/kg of warmed (39 °C) normal saline (0.9%), with 5 min recoveries between lavages until dynamic compliance was 50% of the non-lavaged value or SpO_2_ 90–92% on FiO_2_ of 0.5 [23]. The FiO_2_ was reduced to 0.5 and animals were deemed surfactant-deficient when PaO_2_ < 75 mmHg was confirmed with two consecutive ABGs 30 min apart following lavage/injury. Additional lavages were administered as needed to establish these lavage goals. Baseline oxygenation goals were established based on FiO_2_, PaO_2_, and mean airway pressure (MAP) values previously reported in mechanically ventilated premature infants prior to receiving surfactant [25].

### 2.3. Experimental Protocol

Following lung lavage, animals were allocated to one of 4 groups: (1) Controls (no surfactant, *n* = 2), (2) ‘Direct-Instillation’ with liquid bolus via ETT catheter: 108 mg/kg of body weight (*n* = 5), (3) Low Dose Aerosol at 54 mg/kg (*n* = 5) or (4) High Dose Aerosol at 108 mg/kg (*n* = 5). The Control animals (*n* = 2) data were included to describe the stability of the animal model over time; however, no statistical analysis between Controls and treatment group or comparisons were made. Treatment arm subjects received the bovine surfactant SF-RI 1 (AlveoFact, Lyomark Pharma, Oberhaching, Germany). SF-RI-1 is composed of approximately 90% phospholipids and 1% surfactant protein B and C and is commonly administered to neonates by tracheal liquid instillation. A single vial of powdered SF-RI 1 (108 mg) was reconstituted with 2.4 mL of HCO_3_ buffered saline diluent according to manufacturer instructions (final concentration 45 mg/mL) and body weight-based dosing (108 or 54 mg/kg of body weight) was calculated and administered based on total volume drawn into s syringe (2.4 mL/kg for High Dose Aerosol and Direct-Instillation, 1.2 mL/kg for Low Dose Aerosol). 

### 2.4. Intratracheal Liquid Surfactant Administration (Direct-Instillation)

Liquid bolus surfactant instillation was administered via ETT based on a standard protocol and per manufacturer’s label dose of 108 mg/kg with the head in a midline position during mechanical ventilation [1]. Using a 5 FR Multi Access Catheter (Kimberly-Clark, Dallas, TX, USA) advanced to the distal end of the ETT, liquid surfactant was divided into two separate aliquots with each delivered into one of the two main bronchi by gentle rotation and positioning of the animal with either the right or left side down. Instillation of initial ½ dose of 108 mg/kg was followed by ventilation for several minutes, before changing position to opposite side down and instilling the remaining ½ dose (the total instilled dose: 108 mg/kg). Time was given between doses for the animal to stabilize as needed. 

### 2.5. Breath-Synchronized Aerosolized Surfactant (High and Low Dose Aerosol)

Nebulized surfactant was delivered with a prototype Photo-Defined Aperture Plate (PDAP) VMN composed of a two-layer architecture with approximately 20,000 precision-formed apertures. This allows for increased liquid output while generating small aerosol droplet sizes (<3 µm). The nebulizer synchronized drug delivery based on the signal from a flow sensor in the inspiratory limb of ventilator circuit, to generate aerosol for the first 80% of each inspiration, minimizing drug loss during exhalation. A schematic of the nebulizer system is shown in Figure 2. Following surfactant nebulization, flow sensors were briefly rinsed with ~2 mL of sterile water and recalibrated, as needed. 

### 2.6. Peridosing Events 

Peridosing adverse events were established a priori based on variability in monitored heart rate (±20%), BP (±20%), SpO_2_ (<90%) and peak inspiratory pressure (PIP, >5 cmH_2_O) to evaluate hemodynamic instability, acute hypoxemia, and airway obstruction, respectively. Events were based on at least one occurrence per animal for each parameter in the period immediately prior to and through completion of surfactant administration (time zero). We also noted the effects of liquid and aerosol surfactant on ventilator flow sensor performance, triggering, and ability to provide consistent volume-guarantee ventilation.

### 2.7. Physiologic Measurements 

Gas exchange (ABGs), ventilation and hemodynamic parameters, Pes, EIT, flow and pressure data were obtained at pre-lavage, post-lavage (baseline, pre-treatment condition) and at 30 min intervals following completion of surfactant delivery for 3 h following SRT. The post-lavage condition represents the baseline condition following surfactant depletion and establishment of VILI. The period between baseline and time zero represents the time where liquid or aerosolized surfactant was administered in the treatment groups, with time zero representing completion of treatment. The time increments represent values obtained 30 min after completion of the treatment and at 30 min intervals thereafter (Figure 1).

Respiratory rate, PIP, PEEP, and MAP were processed using MATLAB software. Disease severity calculation was based on ventilator parameters and ABG values at baseline and 30 min intervals as Oxygenation Index (OI), ventilation efficiency index (VEI), alveolar-arterial oxygen gradient (A-aDO_2_), and dynamic compliance (Cdyn) were calculated [26,27,28]. Indices of breathing effort were based on pressure rate product (PRP) which is the product of peak-to-trough change in esophageal pressure (ΔPes, cmH_2_O) × respiratory rate (breaths/minute) generated by animals while breathing during assisted ventilation [23]. EIT measurements were used to estimate changes in end-expiratory lung volume (EELV) loss and recovery related to surfactant administration, as described in more detail elsewhere [29]. End-expiratory lung impedance (EELI) is representative of EELV based on previously established linear relationships between impedance and functional residual capacity (FRC) within the lungs [30,31,32]. Changes in EELI (ΔEELI) were calculated based on changes between baseline (post-lavage) and at 180 min EELI following surfactant delivery.

Animals were euthanized with 100 mg/kg Euthasol following the 180-minute post-surfactant measurements and the ETT was clamped for 3 min to facilitate oxygen absorption and lung collapse. The static pressure-volume relationship was evaluated in degassed lungs by stepwise inflation in 5 mL increments with a calibrated glass syringe from 0 to a maximum value of 25 mL, followed by stepwise deflation. Pressure recordings were made at each step following a 20 s equilibration period. The mean static compliance (ΔV/ΔP) at maximum volume (25 mL) and closing pressures at each of the deflation volumes were evaluated between groups. 

### 2.8. Statistical Analysis 

Statistical analysis was performed on 3 treatment groups. Controls are shown for descriptive purposes but were underpowered and not included in the statistical analysis. Data were calculated as mean ± standard deviation (SD), for all continuous physiologic outcome variables for surfactant treated groups at pre-treatment baseline and every 30 min following SRT during 3 h post-treatment period. Comparison of multiple groups for post-lavage baseline data was carried out by one-way analysis of variance (ANOVA). Two-way ANOVA was used for evaluation of surfactant treatment effects of group and time and their interaction. Tukey’s multiple comparisons test was used to assess mean post hoc differences at baseline and at each time point between surfactant-treated groups. Dunnett’s test was used to compare differences between pre-treatment baseline and at each timepoint following SRT. EIT measurements were not sampled at all timepoints and were prioritized based on available data which included measurements at baseline and 180 min post completion of surfactant and compared using paired *t* test. Peridosing adverse events were reported based on descriptive analysis. The criterion for significance was *p* < 0.05 for all comparisons. 

## 3. Results

### 3.1. Animal Descriptions 

Rabbits weighed 1.51 ± 0.20 kg (*n* = 17). Baseline data following induction of lung lavage for surfactant-deficiency were not different between groups on gas exchange, disease severity and dynamic compliance (Table 1). The SRT duration period was 21 ± 6, 33 ± 7, and 11 ± 4 min for Low Dose Aerosol, High Dose Aerosol, and Direct-Instillation, respectively

### 3.2. Peridosing Events

A total of 12 adverse events were observed in 5 animals in the Direct-Instillation group during surfactant delivery. The details of them were airway obstruction in 3 (60%) animals, acute hypoxemia in 4 (80%), bradycardia in 3 (60%), and hypotension in 2 (30%). Hypertension was observed in one animal (20%) in the High Dose Aerosol group and there were no adverse events in the Low Dose Aerosol group. 

### 3.3. Hemodynamics, Gas exchange and Ventilation Parameters 

There were no differences in mean systemic BP between treatment groups (*p* = 0.15) following SRT. Gas exchange and ventilation parameters are shown in Figure 3. 

There were no differences in pH or PaCO_2_ between treatment groups or within groups and at the different timepoints following treatment. There were improvements in PaO_2_ in surfactant-treated groups 30 min after treatment, whereas PaO_2_ values remained consistently low (~75) throughout the experiment in the Controls. High Dose Direct-Instillation and Aerosol Groups showed significant improvement in PaO_2_ from respective pre-treatment baseline following surfactant, whereas Low Dose Aerosol did not. The High Dose Aerosol group had higher PaO_2_ than Low Dose Aerosol at at150 and 180 min. PaO_2_ did not vary between groups at any other timepoint. The High Dose Aerosol RR was lower at 120, 150, and 180 min when compared to respective baseline and at 180 min when compared to Low Dose Aerosol. PIP was greater with Direct-Instillation than Low Dose Aerosol at 30- and 180 min post SRT.

### 3.4. Indices of Disease Severity 

Indices of Disease Severity are shown in Figure 4. All treatment groups had improved OI, with significant reductions post-treatment when compared to respective baseline condition; whereas OI in Controls remained increased throughout the 3 h. High Dose Aerosol had greater VEI than all treatment groups at 150- and 180 min following SRT and at 30, 150, and 180 min when compared to baseline condition. High Dose Aerosol and Direct-Instillation groups had lower A-a DO_2_ and PRP post-treatment treatment when compared to baseline, but Low Dose Aerosol did not. There were no differences in OI, PRP, and A-aDO_2_ between groups at any timepoint following treatment. 

### 3.5. Electrical Impedance Tomography 

There were no differences in ΔEELI between treatment groups (Figure 5, *p* = 0.3). Relative improvement in alveolar recruitment and recovery from baseline was observed based on increased ΔEELI in both high dose Direct-Instillation and Aerosol groups; whereas Control group showed evidence of alveolar collapse and consolidation at 180 min and Low Dose Aerosol showed no improvement based on the images in Figure 5. 

### 3.6. Lung Mechanics

Dynamic compliance was not different between groups or from baseline within groups following treatment (Figure 4). Post-mortem static compliance and closing pressure at 20 mL was not different between treatment groups (Figure 6). Static compliance values were 1.2 ± 0.02, 1.1 ± 0.04, 1.1 ± 0.09 and 0.62 ± 0.07 mL/cmH_2_O, for Direct-Instillation, High Dose Aerosol, Low Dose Aerosol, and Controls, respectively. 

## 4. Discussion

This study is the first to show equivalent response to administration of the same dose of bovine surfactant by breath-synchronized nebulization with small particle aerosols and direct-instillation, with fewer peridosing complications in the aerosol groups, using a surfactant-deficient, lung injured animal model of severe respiratory distress. The High Dose Aerosol group had a similar oxygenation response from baseline condition as Direct-Instillation but with less disease severity (VEI) than all treatment groups at 2.5 and 3 h.

A majority of animals receiving Direct-Instillation SRT experienced some degree of airway obstruction, acute hypoxemia, and hemodynamic instability during surfactant delivery, whereas aerosol groups did not. Previous findings in pre-term infants showed acute airway obstruction with liquid bolus surfactant that coincided with PIP increase (>5 cmH_2_O) during volume-targeted ventilation [33]. The same report [33] showed a complete cessation of flow down the ETT in 95% of infants receiving liquid surfactant. We observed 60% of animals from Direct-Instillation group with evidence of airway obstruction, which coincided with hemodynamic instability and acute hypoxemia. Although mean PaCO_2_ in these spontaneously breathing animals was not different at 3 h, the Direct-Instillation group remained in acute respiratory failure (53 mmHg) while failure resolved (40 mmHg) in the High-Dose Aerosol group 3 h post SRT. This finding may be associated with aerosolized surfactant causing less airway obstruction than liquid or achieving more homogenous distribution. These aerosol findings highlight a potentially significant advancement in surfactant delivery for mechanically ventilated infants with RDS.

Numerous reports on improvement in gas exchange in animal models upon aerosol delivery of surfactant have raised hopes for potential successful application of aerosolized surfactant in humans [21,34]. However, enthusiasm has been dampened by apparent requirement for aerosol doses that are multiples of effective instilled liquid doses. Using a similar rabbit model of surfactant-deficiency and ventilation as the current study, Fok et al. [22] aerosolized 375 mg bovine surfactant (Survanta) over 1 h with continuous jet and ultrasonic nebulizers but reported poor pulmonary deposition (<1%) with minimal effects on PaO_2_. In contrast, we observed a two-fold increase in PaO_2_ from baseline in as little as 30 min using 1/3 of the surfactant dose previously reported by Fok et al. [22]. This is likely due to greater delivery efficiency of breath-synchronized aerosol with smaller particles, resulting in a concentrated bolus of aerosol delivered for the initial part of inhalation with minimal expiratory drug loss.

Unlike the nebulizer protype used in the current study, which makes aerosol available during inhalation, traditional nebulizers produce aerosol continuously throughout the respiratory cycle. Constant output nebulizers have demonstrated extremely poor lung deposition of aerosolized surfactant with up to 99% of surfactant lost in the expiratory limb of the ventilator tubing, Y piece, and nebulizer [16]. Such high losses necessitate inhaled aerosol doses that are multiples of instilled doses to achieve effect. Jorch and colleagues, using a continuous jet nebulizer to deliver SRF-1, were one of the few reporting a promising clinical response, but concluded that aerosol doses up to 4-fold larger than liquid instillation made aerosol impractical in terms of both cost and time of administration, dampening enthusiasm for inhaled surfactant with aerosol delivery options then available [21]. As such, limitations of these aerosol delivery systems have rendered clinical application to infants impractical for SRT. 

We anticipated similar outcomes between direct-instillation and aerosol at the same surfactant dose (108 mg/kg) based on high efficiency aerosol output data from preliminary studies in vitro (data not shown). However, since nebulization with small aerosol droplets may have potential for more uniform surfactant distribution in the distal airways than liquid bolus instillation, we speculated that low dose aerosolized surfactant (54 mg/kg) may have a similar effect. Interestingly, while PaO_2_ was greater with High Dose Aerosol, effects on disease severity, RR, compliance, and PRP did not differ between Low Dose Aerosol and Direct-Instillation, and still there were no peridosing events with aerosol. 

We did not observe improvements in dynamic compliance between surfactant groups or controls. This finding is consistent with studies in human neonates, where surfactant administration failed to demonstrate early improvement of compliance [35]. Spontaneously breathing animals from the current study were actively generating pleural pressures in combination with assisted ventilator breaths, ventilation; both have been shown to overestimate dynamic lung compliance because proximal airway and alveolar pressures never truly equilibrate at end-inhalation. However, post-mortem static pressure volume curves showed a nearly two-fold greater lung compliance in surfactant-treated rabbits than did controls and were clearly more representative of pulmonary mechanical properties than dynamic compliance. We observed greater stabilization and recruitment with surfactant than controls at 20 mL deflation, a volume similar to the FRC of a pre-term infant [10]. Alveolar surface tension due to surfactant-deficiency and lung collapse in the expiratory phase reduces the functional surface area for gas exchange to occur, places infants at a greater risk for lung injury, and requires higher transpulmonary pressure and work of breathing (WOB) to maintain lung inflation [5]. While we did not show any differences in EIT, based on highly variable breathing efforts, descriptive ΔEELI measurements support a trend toward higher EELV and FRC which coincide with improvements in oxygenation response using a standard label dose for surfactant with direct-instillation and aerosolized surfactant. Our findings provide an empiric demonstration that aerosolized surfactant may be efficacious, efficient, and safe, but clinical studies are needed to determine subsequent dosing and ability to wean preterm human infants more rapidly from invasive mechanical ventilation. 

Since the prototype nebulizer device emitted a small particle aerosol during inhalation, aerosol is delivered to the lungs with minimal aerosol impaction or cooling effects on proximal hot-wire flow sensor performance and volume-guarantee ventilation during aerosol treatment. During dosing in the initial High Dose Aerosol animal, brief hypertension was observed with a small reduction in PIP and tachypnea during nebulization. A small amount of residual surfactant was noted to have accumulated on the proximal flow sensor towards the end of surfactant delivery to this animal. These effects were not considered to be clinically important and were quickly alleviated by rinsing and recalibrating the flow sensor when the nebulizer was removed following treatment. These effects were overcome in subsequent animals by adjusting the aerosol output spray time to prevent retrograde movement of aerosol into the flow sensor at end-inhalation. 

### 4.1. Limitations

The major limitations of the present study derive from the use of an animal model with different airway and pulmonary anatomies than are found in prematurely born human infants, along with mature lung structure undergoing active surfactant metabolism. We included a limited number of controls (*n* = 2) to provide descriptive data to show rate of endogenous surfactant repletion and recovery against treatment groups. Unlike other studies [23,36], that also showed lavage maintains surfactant-deficiency over several hours of continued ventilator support, Controls were underpowered to do a comparative statistical analysis with treatment groups. Additionally, since these animals were relatively small, sedated, and had severe lung disease, it is difficult to extrapolate from these animal findings to human pre-term newborns with RDS, especially if they are larger, have a milder form of lung disease, or are not sedated. This pre-clinical study on dosing and safety was conducted prior to evaluating aerosol surfactant administration via nasal continuous positive airway pressure (CPAP) in human pre-term infants. Since there have been no studies using breath-synchronized VMNs in small animal models, we chose to initially study intubated and fully supported rabbits to safely compare aerosol to standard bolus SRT. We used female rabbits in this study because female is biologically the stronger sex and can tolerate the manipulation for RDS model. 

### 4.2. Clinical Implications for the Proposed Research

This preclinical model suggests that inhaled surfactant at similar or lower than instilled doses may provide a more cost-effective alternative for liquid bolus instillation, possibly with lower risk to patients. Breath-synchronized VMN with PDAP has potential to be a safe, effective and economical alternative to direct instillation. This therapy could also provide an alternative to intubation and mechanical ventilation for surfactant delivery in settings that lack the necessary resources to provide intubation or catheter placement for bolus instillation, reducing delays in providing treatment, and potentially reducing need for transport to centers with such resources. 

Our findings of response to inhaled dose may not be directly applicable to spontaneously breathing animals or human infants receiving nasal CPAP. Avoiding intubation and ventilation solely for the purpose of delivering surfactant could be a breakthrough for surfactant delivery and lung protection, assuming drug delivery can be accomplished with this novel technology. These findings were critical to the decision to advance this delivery technology to multi-center clinical trials currently in progress for preterm infants supported by nasal CPAP. 

## 5. Conclusions

Breath-synchronized VMN has potential as a safe, effective and economical alternative to bolus instillation of surfactant. Aerosol provided a safe and effective alternative to direct-instillation during mechanical ventilation in this animal model. Studies evaluating physiologic outcomes and safety using Breath-synchronized VMN with noninvasive support are currently underway. 

## Figures and Tables

**Figure 1 pharmaceutics-13-01580-f001:**
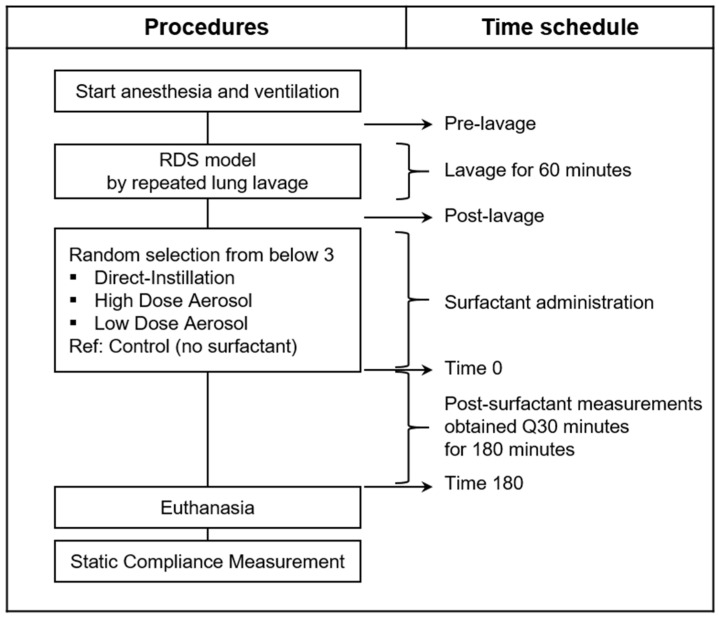
Time schedule for animal experimental procedures. The sequence of procedures for lung lavage to establish surfactant-deficiency and lung injury, randomized groups, and time schedule for measurements. The post-lavage condition shows animals following lung injury and stabilization and represents the baseline pre-treatment condition and the time at which treatment was instituted. Time point 0 represents timepoint following completion of surfactant administration. The time increments represent values obtained 30 min after completion of treatment and at 30 min intervals thereafter.

**Figure 2 pharmaceutics-13-01580-f002:**
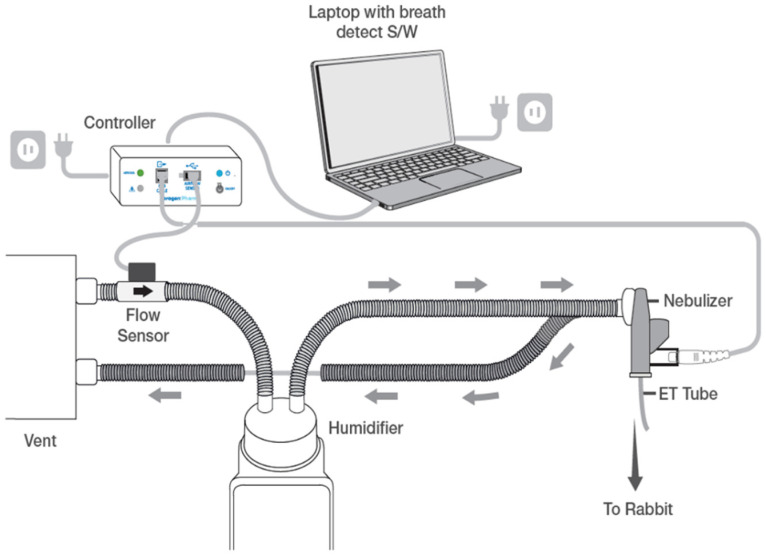
Nebulizer system. Schematic of the surfactant delivery system used for animals. This nebulizer produces aerosol with a novel, two-layer vibrating mesh aperture plate and prevents drug loss to the system on exhalation by synchronizing drug delivery to a ventilator breath by means of a flow change sensor placed within the inspiratory limb of the ventilator circuit to detect inspirations and provide signal to a controller. Controller sends signal to vibrating mesh nebulizer placed between vent circuit and endotracheal tube to emit aerosol for the first 80% of each breath. (Schematic provided courtesy of Aerogen Pharma).

**Figure 3 pharmaceutics-13-01580-f003:**
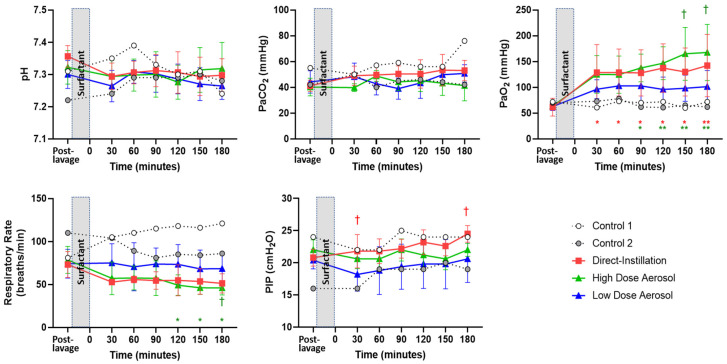
Gas exchange and Ventilation Parameters. Data are represented as individual data for Control and mean ± SD for Surfactant treatment groups during the 180 min treatment period. The post-lavage condition shows animals following lung injury and stabilization and represents the baseline pre-treatment condition and the time at which treatment was instituted. Time point 0 represents timepoint following completion of surfactant administration. The time increments represent values obtained 30 min after completion of treatment and at 30 min intervals thereafter. Controls are shown for descriptive purposes but were underpowered and not included in the statistical analysis. There were no differences in pH or PaCO_2_ between treatment groups or within groups between the respective baseline pre-treatment values and at the different timepoints following treatment. There were improvements in PaO_2_ in surfactant-treated groups 30 min after treatment, whereas, PaO_2_ values remained consistently low (~75) throughout the experiment in the Controls. High Dose Direct-Instillation and Aerosol Groups showed significant improvement in PaO_2_ from baseline following surfactant; whereas, Low Dose Aerosol did not. The High Dose Aerosol group had higher PaO_2_ than Low Dose Aerosol at at150 and 180 min. PaO_2_ did not vary between groups at any other timepoint. The High Dose Aerosol RR was lower at 120, 150, and 180 min when compared to respective baseline and at 180 min when compared to Low Dose Aerosol. PIP was greater with Direct-Instillation than Low Dose Aerosol at 30- and 180 min post surfactant replacement therapy. *, *p* < 0.05; **, *p* < 0.01 vs. baseline. †, *p* < 0.05 vs. Low Dose Aerosol. PaCO_2_, partial pressure of carbon dioxide in the arterial blood; PaO_2_, partial pressure of oxygen in the arterial blood; PIP, peak inspiratory pressure.

**Figure 4 pharmaceutics-13-01580-f004:**
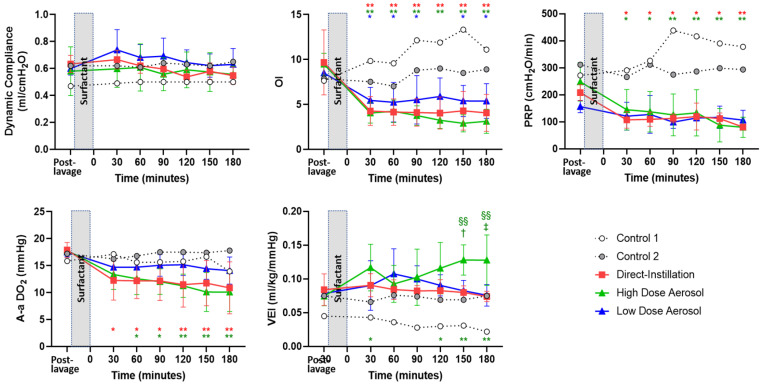
Disease Severity and Effort of Breathing. Physiologic data are represented as individual data for Control and mean ± SD for Surfactant treatment groups during the 180 min treatment period. The post-lavage condition shows animals following lung injury and stabilization and represents the baseline pre-treatment condition and the time at which treatment was instituted. Time point 0 represents timepoint following completion of surfactant administration. The time increments represent values obtained 30 min after completion of treatment and at 30 min intervals thereafter. Dynamic compliance was not different between groups or from baseline following treatment. All treatment groups had improved OI, with significant reductions post-treatment when compared to respective baseline condition; whereas OI in Controls remained increased throughout the 3 h. High Dose Aerosol had greater VEI than all treatment groups at 150- and 180 min following surfactant replacement therapy and at 30, 150, and 180 min when compared to baseline condition. High Dose Aerosol and Direct-Instillation groups had lower A-a DO_2_ and PRP post-treatment treatment when compared to baseline but Low Dose Aerosol did not. There were no differences in OI, PRP, and A-aDO_2_ between groups at any timepoint following treatment. *, *p* < 0.05; **, *p* < 0.01 vs. baseline. †, *p* < 0.05; ‡, *p* < 0.01 vs. Low Dose Aerosol. §§, *p* < 0.01 vs. Direct-Instillation. OI, oxygenation index; VEI, ventilation efficiency index. A-a DO_2_, alveolar-arterial difference in oxygen tension; PRP, pressure rate product.

**Figure 5 pharmaceutics-13-01580-f005:**
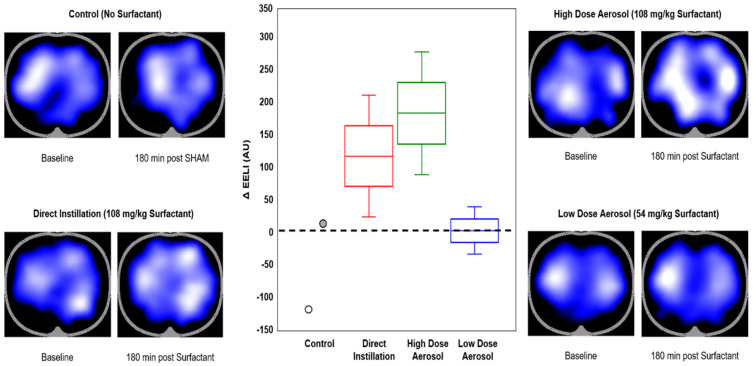
Electrical Impedance Tomography (EIT) and end-expiratory lung volume (∆EELV). Representative EIT images at baseline (post-lavage) and 180 min after intervention with Control (**upper left**), Direct Instillation (**upper right**), High Dose Aerosol (**lower right**) and Low Dose Aerosol (**lower left**). The dark areas of the EIT images represent areas with no ventilation with blue representing areas of increased ventilation distribution with white coloring showing maximal ventilation. Representative images were typical of animals with similar oxygenation response following surfactant replacement. The box (median and interquartile range) and whisker (minimum and maximum ranges) plot depicts relative changes in EELI (Au) between baseline and 180 min condition (**center**). There were no differences in ΔEELI between groups (*p* = 0.3). Relative improvement in alveolar recruitment and recovery from baseline was observed with increased ΔEELI in both high dose Direct-Instillation and Aerosol groups; whereas Control group showed evidence of alveolar collapse and consolidation over 180 min and Low Dose Aerosol showed no improvement.

**Figure 6 pharmaceutics-13-01580-f006:**
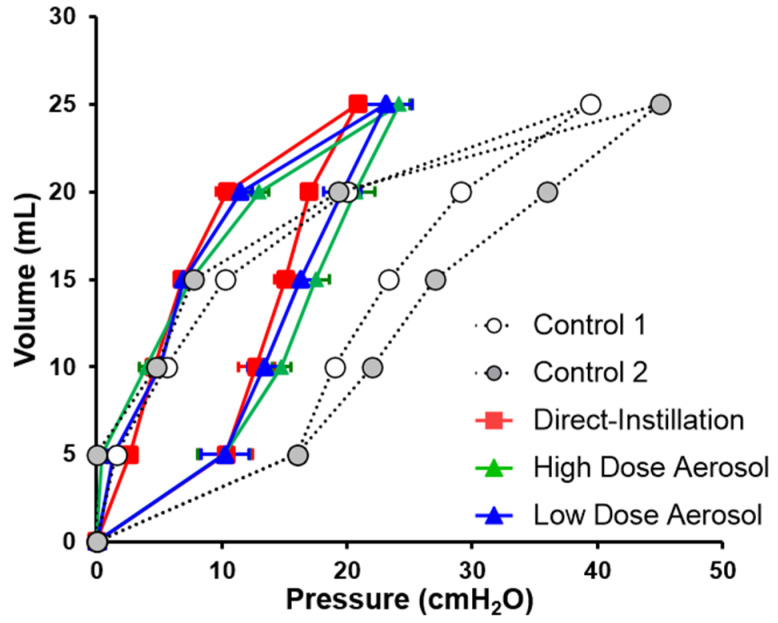
Post-mortem inflation and deflation curves. Pressure values shown are means ± SD for inspiratory and expiratory volumes (0–25 mL). Static compliance was calculated based on mean values at for separate conditions based on a linear fit model. Static compliance values were 1.2 ± 0.02, 1.1 ± 0.04, and 1.1 ± 0.09 mL/cmH_2_O, for Direct-Instillation, High Dose Aerosol, and Low Dose Aerosol, respectively. Compliance and closing pressures at 20 mL (FRC) were not different between treatment groups. Controls are shown for descriptive purposes but were underpowered and not included in the statistical analysis.

**Table 1 pharmaceutics-13-01580-t001:** Post-lavage Baseline (Pre-Treatment).

	Control (*n* = 2)	Direct-Instillation (*n* = 5)	High Dose Aerosol (*n* = 5)	Low Dose Aerosol (*n* = 5)	*p* Value
BW (kg)	1.25, 1.55	1.57 ± 0.17	1.54 ± 0.24	1.47 ± 0.20	0.75
PaCO_2_ (mmHg)	55, 42	41 ± 4	40 ± 7	44 ± 8	0.64
PaO_2_ (mmHg)	72, 70	62 ± 17	64 ± 7	63 ± 8	0.95
Compliance (ml/cmH_2_O)	0.47, 0.62	0.63 ± 0.06	0.58 ± 0.18	0.60 ± 0.05	0.76
OI	7.64, 7.88	9.68 ± 3.61	9.48 ± 1.21	8.51 ± 0.90	0.69
VEI (mL/kg/mmHg)	0.05, 0.08	0.08 ± 0.02	0.07 ± 0.01	0.08 ± 0.01	0.66
A-aDO_2_ (mmHg)	15.9, 17.2	17.9 ± 1.4	17.8 ± 0.6	17.5 ± 0.9	0.84
Lavages (n)	10, 9	10 ± 2	10 ± 2	9 ± 3	0.92

Post-lavage variables are shown following induction of surfactant-deficiency and lung injury. All ventilated animals were supported on FiO_2_ of 0.5 over. BW, body weight; PaCO_2_, partial pressure of carbon dioxide in the arterial blood; PaO_2_, partial pressure of oxygen in the arterial blood; OI, oxygenation index; VEI, ventilation efficiency index; A-aDO_2_, alveolar to arterial oxygenation difference. *p* values were compared among 3 treatment groups (Direct-Instillation, High Dose Aerosol, and low Dose Aerosol) using one-way ANOVA. Values in Control group are shown as individual data and in other groups as mean ± SD.

## Data Availability

The data presented in this study are available from the corresponding author on reasonable request.

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
