# Peer review of "Physiologic Effects of Instilled and Aerosolized Surfactant Using a Breath-Synchronized Nebulizer on Surfactant-Deficient Rabbits [Author-notes fn1-pharmaceutics-13-01580]"

_pharmaceutics, 2021, doi:10.3390/pharmaceutics13101580_

Round 1
Reviewer 1 Report
The overall manuscript is not written in the style of the Journal, it is not well organized and these type of results are more appropriate for another type of Journal
Author Response
We appreciate the reviewer’s careful critique and the many important suggestions regarding our manuscript:pharmaceutics-1281996; PHYSIOLOGIC EFFECTS OF INSTILLED AND AEROSOLIZED SURFACTANT USING A BREATH-SYNCHRONIZED NEBULIZER ON SURFACTANT-DEFICIENT RABBITS
Our responses, which we hope are satisfactory, are below.
Reviewer #1
The overall manuscript is not written in the style of the Journal, it is not well organized and these type of results are more appropriate for another type of Journal.
Authors Response: We respectfully disagree. Development of drugs to premature very low weight infants represents an important frontier for pharma. This study represents an important step in progressing to effective methods for aerosol dosing to this population.
Reviewer 2 Report
The manuscript entitled “Physiologic Effects Of Instilled And Aerosolized Surfactant Using A Breath-Synchronized Nebulizer On Surfactant-Deficient Rabbits” deals with the aerosolization of pulmonary surfactant using a novel breath-actuated vibrating mesh nebulizer, compared to the direct-instillation of liquid surfactant via an endotracheal tube.
This topic is of high interest and experimental studies affording this issue are beneficial to determine whether aerosolization is advantageous or not. The paper is well written and the results are interesting. Nevertheless, there are some aspects that could be improved
Comments and Suggestions for Authors
Introduction
The introduction does not provide sufficient background and relevant references.
A recent article (Brasher M., et al Children 2021, 8, 493) presents a historical narrative spanning sixty years of development of aerosolization systems for pulmonary surfactant. Experimental and clinical trials have been carried out to study the advantages of surfactant nebulization. A summary should be provided in the introduction, and the reseach done in this study must be justified on the basis of previous studies and findings on surfactant nebulization.
Methods
- Sample size calculation using data from previous study may be This must be confirmed/validated with results from the current study
- Why 2 animal in control group?
- Why female rabbits?
- Statistical power of results considering the corresponing sample size (2 for control group, 5 or 12? for bolus instilation, 5 for nebulization) ? Please, provide information
Surgical preparation/instrumentation
- Pg 7, second paragraph “The respiratory rate was adjusted initially to maintain pH of 7.35 -7.45 and remained at this setting…” pH of what? Please explain
Experimental protocol
- Pg 9, first paragraph, last sentence “A single vial of powdered SF-RI 1 (108 mg) was reconstituted with HCO3 buffered saline diluent and dosing (mg/mL) was calculated and administered based on total volume drawn into s syringe”. Whic was the total volume of HCO3 buffered saline used for dilution o vial content? Please, provide this
- What is dosing (mg/mL)? Dose units are mass units (mg, µg, ng…).The amount of surfactant and the dilution volumen must be provided
- Pg 9, second paragraph “Instillation of ½ dose was followed by ventilation for several minutes, before changing position to opposite side down and instilling the remaining ½ dose” Instilated volume? Wasted product?
Intratracheal Liquid Surfactant Administration
- Volume of liquid bolus instilated?
The way of administration is essential for delivery, distribution, deposition, and dispersion of the surfactant in the lungs. This experimental procedure must be carefully designed and monitored to ensure dose uniformity. The authors must explain in detail this issue
- Since surfactant deposits in circuit and endotracheal tube, only a fraction of dose delivered deposits in the lung (Lung Dose). How was the Lung Dose monitored?
Results
- Pg 13, paragraph below the table: “The Direct-Instillation group had 3 (60%) animals with airway obstruction, 4 (80%) with acute hypoxemia, 3 (60%) with bradycardia, and 2 (30%) with hypotension during surfactant delivery. Acordingly, 3+4+3+2 = 12 animals in the direct instilation group, which disagrees with Pg 6, first paragraph: “Final sample size was determined as five animals per treatment group and two descriptive controls” Also disagree with size sample given in table 1
- Since n= 2 for control group, standard deviation must be removed in figures and table. Instead, the two values can be given for this group
- In the dicussion, previous findings on surfactant nebulization may be commented and compared to the present study data.
All abbreviations used must be defined. For instance: FiO2 and PaO2 are not defined
Author Response
We appreciate the reviewer’s careful critique and the many important suggestions regarding our manuscript:pharmaceutics-1281996; PHYSIOLOGIC EFFECTS OF INSTILLED AND AEROSOLIZED SURFACTANT USING A BREATH-SYNCHRONIZED NEBULIZER ON SURFACTANT-DEFICIENT RABBITS
Our responses, which we hope are satisfactory, are below.
Reviewer #2
The manuscript entitled “Physiologic Effects Of Instilled And Aerosolized Surfactant Using A Breath-Synchronized Nebulizer On Surfactant-Deficient Rabbits” deals with the aerosolization of pulmonary surfactant using a novel breath-actuated vibrating mesh nebulizer, compared to the direct-instillation of liquid surfactant via an endotracheal tube.
This topic is of high interest and experimental studies affording this issue are beneficial to determine whether aerosolization is advantageous or not. The paper is well written and the results are interesting. Nevertheless, there are some aspects that could be improved
Authors Response: We thank the reviewer for his/her kind comments.
Comments and Suggestions for Authors
Introduction
The introduction does not provide sufficient background and relevant references.
A recent article (Brasher M., et al Children 2021, 8, 493) presents a historical narrative spanning sixty years of development of aerosolization systems for pulmonary surfactant. Experimental and clinical trials have been carried out to study the advantages of surfactant nebulization. A summary should be provided in the introduction, and the reseach done in this study must be justified on the basis of previous studies and findings on surfactant nebulization.
Authors Response: We thank the reviewer and have added the review reference (Brasher M, Raffay TM, Cunningham MD, Abu Jawdeh EG. Aerosolized Surfactant for Preterm Infants with Respiratory Distress Syndrome. Children (Basel). 2021 Jun 10;8(6):493.) and modified the introduction to describe the relevant history of preclinical animal studies with aerosol surfactant.
Methods
Sample size calculation using data from previous study may be This must be confirmed/validated with results from the current study
Authors Response: Our analysis of results supports the sample size calculation.
Why 2 animal in control group?
Authors Response: As stated in introduction, the control n=2 was selected to show model stability, and not for purposes of comparison between active treatment arms.
Why female rabbits?
Authors Response: Male rabbits are potentially the weaker of the species. Rabbits are well known to have high mortality associated with the manipulations required to prep for the study. Female rabbits seem to better tolerate this work. We included this in the limitation session.
Statistical power of results considering the corresponing sample size (2 for control group, 5 or 12? for bolus instilation, 5 for nebulization) ?
Authors Response: Sample size is n=5 per group (Direct-Instillation, Low Dose Aerosol, and High Dose Aerosol). Additionally the control n=2 was selected to show model stability. Final sample size was determined as five animals per treatment group and two descriptive controls. Statistical analysis was performed on 3 treatment groups. Controls are shown for descriptive purposes but were underpowered and not included in the statistical analysis. (Added in Methods)
12 was total adverse events number in Direct-Instillation group (n = 5). (We have now clearly explained in Result)
Surgical preparation/instrumentation
Pg 7, second paragraph “The respiratory rate was adjusted initially to maintain pH of 7.35 -7.45 and remained at this setting…” pH of what? Please explain
Authors Response: We clarified that this was pH of arterial blood. ,
Experimental protocol
Pg 9, first paragraph, last sentence “A single vial of powdered SF-RI 1 (108 mg) was reconstituted with HCO3 buffered saline diluent and dosing (mg/mL) was calculated and administered based on total volume drawn intos syringe”. Whic was the total volume of HCO3 buffered saline used for dilution o vial content? Please, provide this
Authors Response: The prefilled diluent syringe delivers 2.4 mL to mix with the 108 mg powder cake in the vial. This is the standard reconstitution of the Alveofact SF-RI 1 surfactant as marketed in 28 countries over the last 20+ years.
What is dosing (mg/mL)? Dose units are mass units (mg, µg, ng…).The amount of surfactant and the dilution volume must be provided
Authors Response: Dosing was weight based for each animal at mg/kg. The volume of surfactant instilled or placed in the nebulizer reservoir based on 45 mg/mL.. for example, 1 kg animals received 2.4 mL of surfactant (108 mg) or the lower dose of 1.2 mL (54 mg). (We have now clearly defined in Methods)
Pg 9, second pargraph “Instillation of ½ dose was followed by ventilation for several minutes, before changing position to opposite side down and instilling the remaining ½ dose” Instilated volume? Wasted product?
Intratracheal Liquid Surfactant Administration
Volume of liquid bolus instilated?
Authors Response: The calculated instilled dose was administered per label provided by the manufacturer. This consists of instilling the first 50% of dose, ventilating via the airway ,changing child/animal position and instilling the remaining 50% of the dose. (We have now clearly defined in
Methods)
The way of administration is essential for delivery, distribution, deposition, and dispersion of the surfactant in the lungs. This experimental procedure must be carefully designed and monitored to ensure dose uniformity. The authors must explain in detail this issue
Authors Response: We agree with the reviewer. We designed the protocol to provide precise dosing via both instillation and aerosol as described in our methods. Bolus instillation was based on standard techniques use with intubated preterm infants. Aerosol was administered with the price dose volume placed into the nebulizer and administered until end of aerosol being emitted. This study was defined to quantify effect of aerosol administration in this previously described washout model and demonstrated similar response to same dose by instillation and aerosol. Quantification of delivery, distribution, deposition, and dispersion of the surfactant in the lung was beyond the scope of this study. Future studies with radiolabeled aerosols would be required and currently in our plan as we have expanded lab capabilities to include imaging.
Since surfactant deposits in circuit and endotracheal tube, only a fraction of dose delivered deposits in the lung (Lung Dose). How was the Lung Dose monitored?
Authors Response: We agree. The question is whether the amount that reaches the lung parenchyma is sufficient to elicit the desired effects. There is also a cost of surfactant distribution with instillation airways being coated before reaching the alveoli. This cost is why many have hypothesized that a smaller dose of aerosol entering the lungs may have a greater loss than aerosol that has entered the lungs. We used in vitro models to identify the inhaled dose distal to the trachea with the system used in this paper. Inhaled dose via endotracheal tube ranged from 35-50% depending on ventilatory parameters tested. We evaluated physiologic effects in this study. The other in vitro findings were not from our animal lab.
Results
Pg 13, paragraph below the table: “The Direct-Instillation group had 3 (60%) animals with airway obstruction, 4 (80%) with acute hypoxemia, 3 (60%) with bradycardia, and 2 (30%) with hypotension during surfactant delivery. Acordingly, 3+4+3+2 = 12 animals in the direct instilation group, which disagrees with Pg 6, first paragraph: “Final sample size was determined as five animals per treatment group and two descriptive controls” Also disagree with size sample given in table 1
Authors Response: We thank the reviewer and have clarified the statement.
Since n= 2 for control group, standard deviation must be removed in figures and table. Instead, the two values can be given for this group
Authors Response: We have removed the SD and provided a line of identify for each of the controls.
In the Discussion, previous findings on surfactant nebulization may be commented and compared to the present study data.
Authors Response: We have reviewed the discussion and assure that relevant studies were compared.
All abbreviations used must be defined. For instance: FiO2 and PaO2 are not defined
Authors Response: We thank the reviewer. We have reviewed the manuscript to make sure all abbreviations are defined with table or first use.
Reviewer 3 Report
The authors have carried out an interesting study on the safety and efficacy of surfactant delivery by comparing liquid bolus instillation via ETT with that of aerosol delivery using a breath-synchronized Nebulizer. The results from this study will be very useful for designing human studies. I feel the results and discussions are presented well and with a few minor improvements, the paper can be accepted.
- The Sample size seems to be small. Although it is briefly mentioned in the methods section, please elaborate in detail on how the sample was chosen in the study.
- In table 1, how p values were calculated?
- Although I found the rationale behind including the low-dose aerosol delivery in discussions, please include it in the introduction.
- On page 14, “Gas exchange and ventilation parameters are shown in Figure 2.” It should be figure 3. Similarly, page 18, line 2- figure 3 should be figure 6.
- It is stated that only 1% of the surfactant delivered to the targeted region in previous studies with nebulized delivery. Is there any quantitative data on delivery efficiency with a breath-synchronized Nebulizer (perhaps, in vitro)?
- Since this study only looks at safety and efficacy, how can the current pre-clinical methods be extended to study surfactant delivery efficiency between the two strategies?
Author Response
Reviewer #3
The authors have carried out an interesting study on the safety and efficacy of surfactant delivery by comparing liquid bolus instillation via ETT with that of aerosol delivery using a breath-synchronized Nebulizer. The results from this study will be very useful for designing human studies. I feel the results and discussions are presented well and with a few minor improvements, the paper can be accepted.
Authors Response: We really appreciate the reviewer’s complementary comments on our work.
- The Sample size seems to be small. Although it is briefly mentioned in the methods section, please elaborate in detail on how the sample was chosen in the study.
Author Response: We mentioned in Methods sample size calculation for surfactant treatment (n = 5 per group) was based on power analysis using mean and standard deviation of partial pressure of oxygen in the arterial blood (PaO2) from a previous rabbit study that compared nebulized to directly instilled surfactant in intubated mechanically ventilated rabbits. Final sample size was determined as five animals per treatment group. We included a small and limited number of descriptive control animals (n = 2) that did not receive surfactant to provide descriptive data to show rate of endogenous surfactant repletion and recovery against treatment groups. These animals were not included in the statistical analysis.
- In table 1, how p values were calculated?
Author Response: Great point. Thank You. We included the following in the Table 1 legend: “P values were compared among 3 treatment groups (Direct-Instillation, High Dose Aerosol, and low Dose Aerosol) using one-way ANOVA. Values in Control group are shown as individual data and in other groups as mean ± SD.”
Although I found the rationale behind including the low-dose aerosol delivery in discussions, please include it in the introduction.
Author Response: Thank you for mentioning this. We included the following statement in the Introduction: “In addition to standard treatment doses used with aerosol and direct-instillation, we nebulized 1/2 standard dose in one group to establish dosing between small particles generated with aerosol and the relatively large liquid particles with direct-instillation.”
- On page 14, “Gas exchange and ventilation parameters are shown in Figure 2.” It should be figure 3. Similarly, page 18, line 2- figure 3 should be figure 6.
Author Response: Thank You. We made these necessary changes
- It is stated that only 1% of the surfactant delivered to the targeted region in previous studies with nebulized delivery. Is there any quantitative data on delivery efficiency with a breath-synchronized Nebulizer (perhaps, in vitro)?
Author Response: This is a great question. A separate lab is currently in the process of publishing these in vitro data. This study was specifically designed to evaluate physiologic outcomes. We are excited to see the results upon publication.
- Since this study only looks at safety and efficacy, how can the current pre-clinical methods be extended to study surfactant delivery efficiency between the two strategies?
Author Response: I think this is a great question. Based on our pre-clinical data in animals, we believe that experimental design of clinical studies in humans would consider our safety and efficacy findings. However, dosing in future clinical trials will need to be individualized based on disease severity and response in subjects.
Reviewer 4 Report
Reviewer Report
The article titled “Physiologic Effects Of Instilled And Aerosolized Surfactant Using A Breath-Synchronized Nebulizer On Surfactant-Deficient Rabbits” has been reviewed carefully. In this paper, authors hypothesized that aerosolized surfactant with a novel breath-actuated vibrating mesh nebulizer would have similar efficacy and safety as instilled surfactant.
Recommendation:
The paper is well written and presented. The numerical results, presented through graphs are organized in outstanding way and the outcomes are explained very well in the “Results and Discussion” section. The paper can be published by Pharmaceutics after some revisions.
Comments:
- Did you investigate the effects of properties of the liquids to nebulize?
- Did you compare the efficiency of this nebulizer with other types?
- This work is of interest subject to minor corrections of the minor English errors.
- At several places in the text the word spacing has not been taken care of Insert space after each and every comma and full stop. Typographic errors and word spacing can be fixed in the revised version for running a spell check.
- Please describe Figure 5 more in the text.
- All figure captions and section heads should be uniform.
- Authors can use the suggested work in the revised version.
- Advanced Drug Delivery Reviews, Vol 160, 2020, 105-114
- The reference list in the end is not uniform and it should be corrected as per required style of journal.
Author Response
We appreciate the reviewer’s careful critique and the many important suggestions regarding our manuscript:pharmaceutics-1281996; PHYSIOLOGIC EFFECTS OF INSTILLED AND AEROSOLIZED SURFACTANT USING A BREATH-SYNCHRONIZED NEBULIZER ON SURFACTANT-DEFICIENT RABBITS
Our responses, which we hope are satisfactory, are below.
Reviewer #4
The authors have carried out an interesting study on the safety and efficacy of surfactant delivery by comparing liquid bolus instillation via ETT with that of aerosol delivery using a breath-synchronized Nebulizer. The results from this study will be very useful for designing human studies. I feel the results and discussions are presented well and with a few minor improvements, the paper can be accepted.
Authors Response: We thank the reviewer for the kind and insightful comments.
- The Sample size seems to be small. Although it is briefly mentioned in the methods section, please elaborate in detail on how the sample was chosen in the study.
Authors Response: We did a power analysis
- In table 1, how p values were calculated?
Authors Response: P values were compared among 3 treatment groups (Direct-Instillation, High Dose Aerosol, and low Dose Aerosol) using one-way ANOVA. Values in Control group are shown as individual data and in other groups as mean ± SD. (added in the table legend)
- Although I found the rationale behind including the low-dose aerosol delivery in discussions, please include it in the introduction.
Authors Response: Now it was included in the introduction.
- On page 14, “Gas exchange and ventilation parameters are shown in Figure 2.” It should be figure 3. Similarly, page 18, line 2- figure 3 should be figure 6
Authors Response: Thank you for noticing these errors. We corrected them.
- It is stated that only 1% of the surfactant delivered to the targeted region in previous studies with nebulized delivery. Is there any quantitative data on delivery efficiency with a breath-synchronized Nebulizer (perhaps, in vitro)?
Authors Response: Not in infant models
- Since this study only looks at safety and efficacy, how can the current pre-clinical methods be extended to study surfactant delivery efficiency between the two strategies?
Authors Response: We really appreciate the reviewer’s great question. These findings were sufficient to encourage proceeding to Phase 2b study in 30 preterm infants with RDS. This was also a safety with early indications of efficacy. Delivery efficiency has been addressed for the clinical program with in vitro modeling and will be address in clinical trials with dose ranging. However, we plan to use our new CT scan with radiolabeled administration with various animal models to better understand elements of delivery efficiency in vivo.
Round 2
Reviewer 1 Report
I appreciate the authors 'efforts to improve the quality of the article, but I disagree with the authors' response to my comment as this article is not about drug development.
Reviewer 2 Report
As far as I am concerned, the last version is improved and suitable for publication